# Nickel sulfide nanocrystals on nitrogen-doped porous carbon nanotubes with high-efficiency electrocatalysis for room-temperature sodium-sulfur batteries

Zichao Yan[1], Jin Xiao [2], Weihong Lai[1], Li Wang[1], Florian Gebert[1], Yunxiao Wang [1], Qinfen Gu[3], Hui Liu[4], Shu-Lei Chou [1], Huakun Liu [1] & Shi-Xue Dou [1]

Polysulfide dissolution and slow electrochemical kinetics of conversion reactions lead to low utilization of sulfur cathodes that inhibits further development of room-temperature sodium-sulfur batteries. Here we report a multifunctional sulfur host, $NiS_2$ nanocrystals implanted in nitrogen-doped porous carbon nanotubes, which is rationally designed to achieve high polysulfide immobilization and conversion. Attributable to the synergetic effect of physical confinement and chemical bonding, the high electronic conductivity of the matrix, closed porous structure, and polarized additives of the multifunctional sulfur host effectively immobilize polysulfides. Significantly, the electrocatalytic behaviors of the Lewis base matrix and the $NiS_2$ component are clearly evidenced by operando synchrotron X-ray diffraction and density functional theory with strong adsorption of polysulfides and high conversion of soluble polysulfides into insoluble $Na_2S_2/Na_2S$. Thus, the as-obtained sulfur cathodes exhibit excellent performance in room-temperature Na/S batteries.

[1] Institute for Superconducting & Electronic Materials, Australian Institute of Innovative Materials, University of Wollongong, Innovation Campus, Squires Way, North Wollongong, NSW 2500, Australia. [2] School of Science, Hunan University of Technology, Zhuzhou 412007, China. [3] Australian Synchrotron, 800 Blackburn Road, Clayton, VIC 3168, Australia. [4] Institute of New-Energy Materials, School of Materials Science and Engineering, Tianjin University, Tianjin 300072, China. Correspondence and requests for materials should be addressed to Y.W. (email: yunxiao@uow.edu.au) or to H.L. (email: hui_liu@tju.edu.cn) or to S.-L.C. (email: shulei@uow.edu.au)

Low-cost sulfur-based sodium-ion storage has attracted tremendous interest for next-generation electric energy storage systems to meet increasing demands[1–3]. In the 1960s, high-temperature Na–S batteries were commercialized in smart grid stationary storage. Their operating temperature, however, around 300–350 °C, could potentially introduce severe safety issues and lead to $Na_2S_3$ as the final discharge product with low theoretical energy density of 760 W h $kg^{-1}$ [4,5]. Consequently, room temperature sodium-sulfur (RT-Na/S) batteries are inspiring great interest, which could well address the safety hazard. They exhibit an increased energy density, up to 1274 W h $kg^{-1}$, with $Na_2S$ as the final discharge product. This battery system suffers from rapid capacity fading and low reversible capacity, however, which can be mainly attributed to the sluggish reaction kinetics of sulfur and its $Na_2S$ product, along with serious polysulfide migration[6–10]. Significantly, various sulfur hosts have been developed for Li/S batteries to cope with the similar challenges, including a series of carbon matrices[11–18], and polar sulfur hosts[19–26]. Nevertheless, sulfiphilic sulfur hosts have much lower conductivity than carbon materials, which inevitably compromise the rate capability and specific capacity of sulfur. To date, only a few sulfur hosts have been explored to enable RT-Na/S batteries[4,5,27–37]. By virtue of physical confinement, interconnected hollow mesoporous carbon can effectively encapsulate sulfur species inside of carbon shells during charge/discharge process[27], although the low reversible capacity and insufficient lifespan of the cathode indicate that physical confinement alone is not sufficient to address the soluble polysulfide problem. Thus, constructing a multifunctional sulfur host by coupling a polar component with a functional carbon matrix is a promising way to achieve advancement on RT-Na/S batteries.

Herein, we present a multifunctional sulfur host with $NiS_2$ nanocrystals implanted in nitrogen-doped porous carbon nanotubes ($NiS_2$@NPCTs). First, the one-dimensional conductive NPCTs with a continuous carbon backbone inside can provide short ion diffusion paths and a fast transfer rate. Second, abundant cavities in each porous nanotube can serve as closed containers for sulfur species, guarantying sufficient space for sulfur volumetric expansion and efficient polysulfide containment. Moreover, the implanted $NiS_2$ nanocrystals have a polar feature that can bind strongly to sulfur species and spatially localize the deposition of the sulfide species. Significantly, N-doping sites and the $NiS_2$ polar surface are capable of enhancing the adsorption energy of polysulfides, leading to strong catalytic activity towards polysulfide oxidation.

## Results

**Material characterization**. The $NiS_2$@NPCTs/S nanocomposite with uniform one-dimensional (1D) morphology and nanocrystals encapsulated in a unique structure is prepared by a simplified synthesis strategy (Supplementary Figs. 1–3). As shown in Fig. 1a, the porous structures are well identified by scanning transmission electron microscopy (STEM); the corresponding energy dispersive spectroscopy (STEM-EDS) mapping images show the homogeneous distribution of N and S elements along C backbones. It is noticeable that the $NiS_2$ nanocrystals (average size of about 8.3 nm) are well embedded into the carbon matrix and even the interior void space, which account for 10 wt.% in the composite (Supplementary Fig. 4a, b). To realize the mechanism of $NiS_2$ grown within the carbon tubes, a capillary effect via vacuum treatment is introduced to drive the raw materials (nickel salt and thioacetamide) into the interior pores. For comparison, a control sample was prepared by conducting the same experiment but without vacuum treatment. As displayed in Supplementary Fig. 1f, most of the $NiS_2$ compounds can be visually observed by

SEM without vacuum treatment, indicating the $NiS_2$ compounds were adsorbed on the exterior of NPCTs. However, no trace of $NiS_2$ compounds is observed on the surface of the $NiS_2$@NPCTs/S nanocomposite prepared by vacuum stirring, indicating the $NiS_2$ nanocrystals grow within the carbon tubes. In addition, the following step of liquid nitrogen coupled with freeze-drying can further lock $NiS_2$ within the carbon tubes, and the particle size can be effectively controlled by those pores and cavities at the same time. The EDS line scanning (Fig. 1b) of individual cavities clearly demonstrates that S is favorably dispersed on the surface of the $NiS_2$ nanocrystals, indicating their sulfiphilic property. Fig. 1c contains a high-resolution transmission electron microscopy (HRTEM) image taken on $NiS_2$@NPCTs/S composite shows that the interplanar distance between adjacent lattice planes is 0.279 nm, corresponding to (200) plane of $NiS_2$. The inset 16 formula unit crystal structure model of pyrite $NiS_2$ along [001] projected direction, which is highly consistent with the matched inverse fast Fourier transform (IFFT) pattern, indicating a high degree of crystallinity of the $NiS_2$. In agreement with the X-ray diffraction (XRD) pattern (Fig. 1d), several intensive peaks are well indexed to pyrite $NiS_2$ (JCPDS No. 89–1495). The low-intensity S peaks of well encapsulated sulfur can be attributed to the reduced size of the sulfur after sulfur loading process, indicate the successful encapsulation of sulfur. The loading mass of S in the $NiS_2$@NPCTs/S composite was determined to be 56% (consistent with the Brunauer-Emmett-Teller (BET) analysis in Supplementary Fig. 4c) (Fig. 1e), which is 47% in NPCTs/S, further implying the high adsorption energy of S on $NiS_2$. The slight weight loss of $NiS_2$@NPCTs/S composite at high temperature is attributed to the decomposition of $NiS_2$[36,37]. The X-ray photoelectron spectroscopy (XPS) survey spectrum of the $NiS_2$@NPCTs/S (Supplementary Fig. 5a) shows five characteristic peaks corresponding to S 2p, C 1s, N 1s, O 1s, and Ni 2p, respectively. The binding energy peaks observed in the Ni 2p spectrum (Fig. 1f) at 859 and 874 eV can be ascribed to the $2p_{3/2}$ and $2p_{1/2}$ of pyrite $NiS_2$[38,39]. Two peaks in the S 2p spectrum (Fig. 1g) at 162.9 and 164.0 eV are assigned to the $2p_{3/2}$ and $2p_{1/2}$ orbitals of S in $NiS_2$, while the peaks at 163.4 and 164.7 eV are ascribed to the spin-orbit coupling of S $2p_{3/2}$ and S $2p_{1/2}$ in elemental S. The minor peak at 168.7 eV corresponds to C-$SO_x$ groups[40]. This result suggests the successful encapsulation of active S into the $NiS_2$@NPCTs host. The N 1s spectrum (Fig. 1h) shows the domination of pyridinic and pyrrolic nitrogen at 397.6–399.8 eV[41]. The N-doped carbon could serve as a conductive Lewis base matrix, which is expected to increase the adsorption energy of the polysulfides and promote the conversion kinetics[42]. In the C 1s spectra in Fig. 1i, the three peaks at 288.5, 286.4, and 284.4 eV can be attributed to O–C=O, C–O, and C–C bonds, respectively, for both the $NiS_2$@NPCTs/S and the NPCTs. The C–N bond energy in the $NiS_2$@NPCTs/S (285.2 eV) is slightly lower than that in the NPCTs (285.7 eV), which is likely due to the interaction between C and the loaded S[43,44]. This observation is consistent with the Fourier transform infrared (FTIR) analysis (Supplementary Fig. 5b, c). Surprisingly, the sulfur-impregnated materials exhibit a higher D band to G band intensity ratio ($I_D/I_G$) than the NPCTs (Supplementary Fig. 5d), indicating high inclusion of defect sites on the surface[45–47]. These may provide more active sites for trapping polysulfides.

**Electrochemical investigations of $NiS_2$@NPCTs/S materials**. It is expected that the well-designed nanostructures and critical functional components make $NiS_2$@NPCTs/S a superior cathode for RT-Na/S batteries. It is impressive that $NiS_2$@NPCTs/S delivers the high initial capacity of 960 mA h $g^{-1}$ at 1 A $g^{-1}$, and it maintains a stable capacity of 401 mA h $g^{-1}$ for 750 cycles with

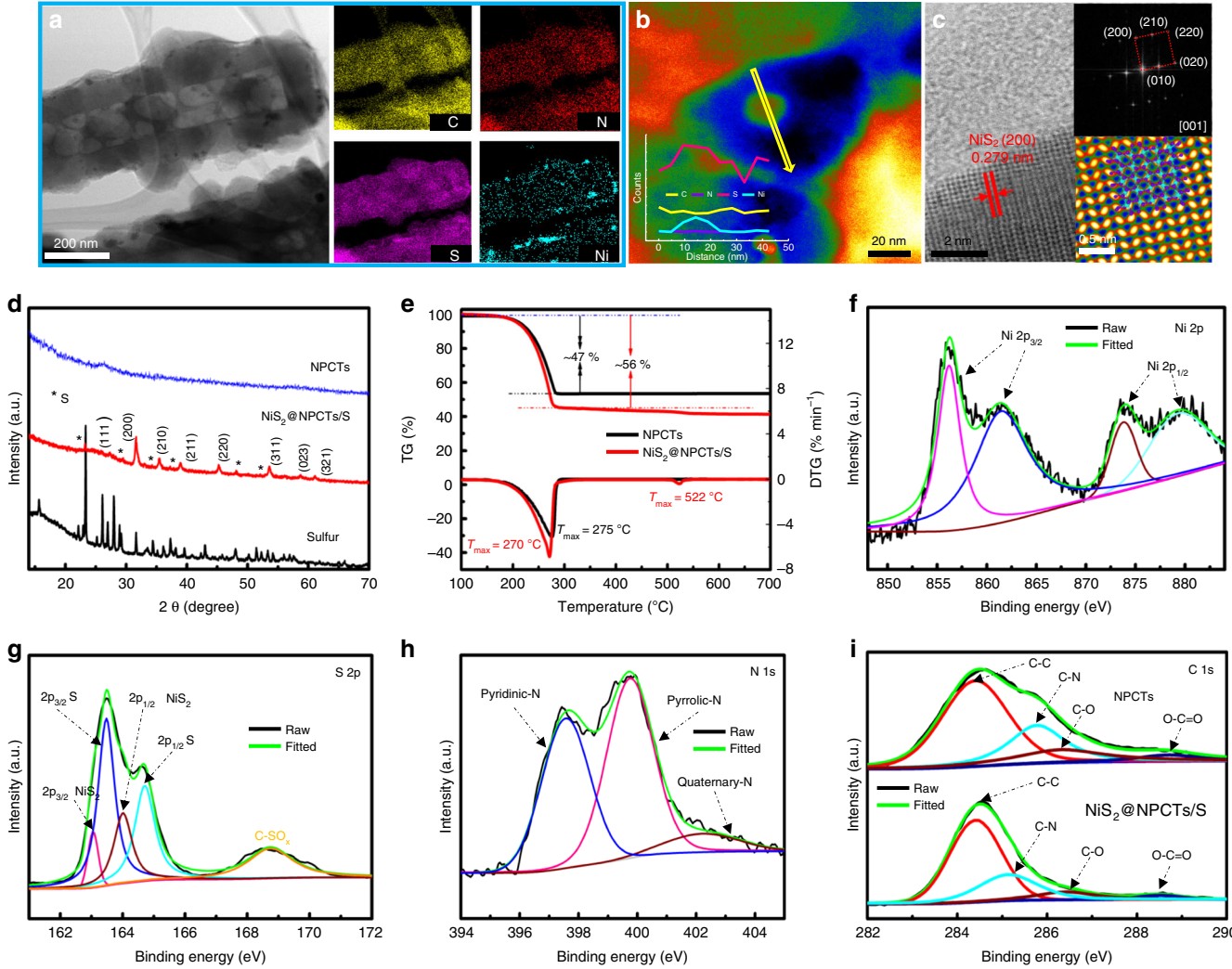

**Fig. 1** Characterizations of as-prepared sample. **a** STEM-EDS mapping images, **b** Colored STEM image coupled with EDS line scanning (inset) of a single cavity, and **c** HRTEM image with corresponding Fast Fourier Transform (FFT) pattern and molecular model matched IFFT image of the NiS$_2$@NPCTs/S composite (insets). **d** XRD patterns of the NPCTs, NiS$_2$@NPCTs/S, and sulfur. **e** Thermogravimetry (TG) and derivative thermogravimetry (DTG) curves of the NPCTs/S and NiS$_2$@NPCTs/S. High-resolution XPS spectra of **f** Ni 2p, **g** S 2p, and **h** N 1s for NiS$_2$@NPCTs/S composite. **i** Comparison of C 1s spectrum between NPCTs and NiS$_2$@NPCTs/S composite

distinct sodiation/desodiation plateaus (Fig. 2a, b), while NPCTs/S shows a large capacity loss of 55% within 100 cycles, highlighting the key role of the NiS$_2$ component. The NiS$_2$@NPCTs/S electrode also exhibits unprecedented rate performance, delivering capacity of 760, 691, 557, 457, 346, and 203 mA h g$^{-1}$ at current density of 0.1, 0.2, 0.5, 1, 2, and 5 A g$^{-1}$, respectively (Fig. 2c). Upon reverting back to 0.1 A g$^{-1}$, the NiS$_2$@NPCTs/S shows a fully restored capacity of 674 mA h g$^{-1}$, which is in good agreement with the reversible capacity of 650 mA h g$^{-1}$ over 200 cycles at 0.1 A g$^{-1}$. Further electrochemical performances are presented in Supplementary Fig. 6. The discharge plateau shown in Fig. 2d can be clearly distinguished even at high rate, indicating the good confinement of sodium polysulfides and the fast reaction kinetics of the NiS$_2$@NPCTs/S electrode. Remarkably, the NiS$_2$@NPCTs/S composite delivered reversible capacity of 327 and 208 mA h g$^{-1}$ for 1800 and 3500 cycles at 2 and 5 A g$^{-1}$, respectively (Supplementary Fig. 7). It is notable that a large irreversible capacity loss is observed in the initial charge/discharge process for both samples, which can be attributed to the surface polysulfide dissolution and irreversible oxidation from polysulfide to sulfur[27,48]. Compared with previous reports, this is the best high-rate cycling stability result for a RT-Na/S battery

with conventional current collector and carbonate-based electrolyte (Supplementary Table 1). In order to exclude the capacity contribution and highlight the advantages of the S host, the electrochemical performances of the NiS$_2$@NPCTs and a commercial carbon nanotube/S mixture (CNTs-S) was compared. The CNTs-S mixture with high crystalline of S was found to be inactive (Supplementary Fig. 8 and 9). The Nyquist spectrum of CNTs-S after 10 cycles shows much higher charge transfer resistance ($R_{ct}$) than that of NiS$_2$@NPCTs/S electrode (Supplementary Fig. 9d), which is fitted to be 1628 and 207 Ω, respectively. When the cells are disassembled, the separator of CNTs-S is brown, which is ascribed to the side product of dissolved polysulfide out of CNTs framework. In contrast, no obvious change in the electrode and separator was observed in NiS$_2$@NPCTs/S electrodes (Supplementary Fig. 10). Moreover, the SEM and cross-profile EDS mapping images of cycled CNTs-S electrodes show that thick film is formed on the electrode surface with dramatically reduced signal of sulfur. By contrast, uniform dispersion of S and Na is observed in NiS$_2$@NPCTs/S. Therefore, the severe polysulfides dissolution and formation of thick passivation film for CNTs-S lead to its failure in Na–S system.

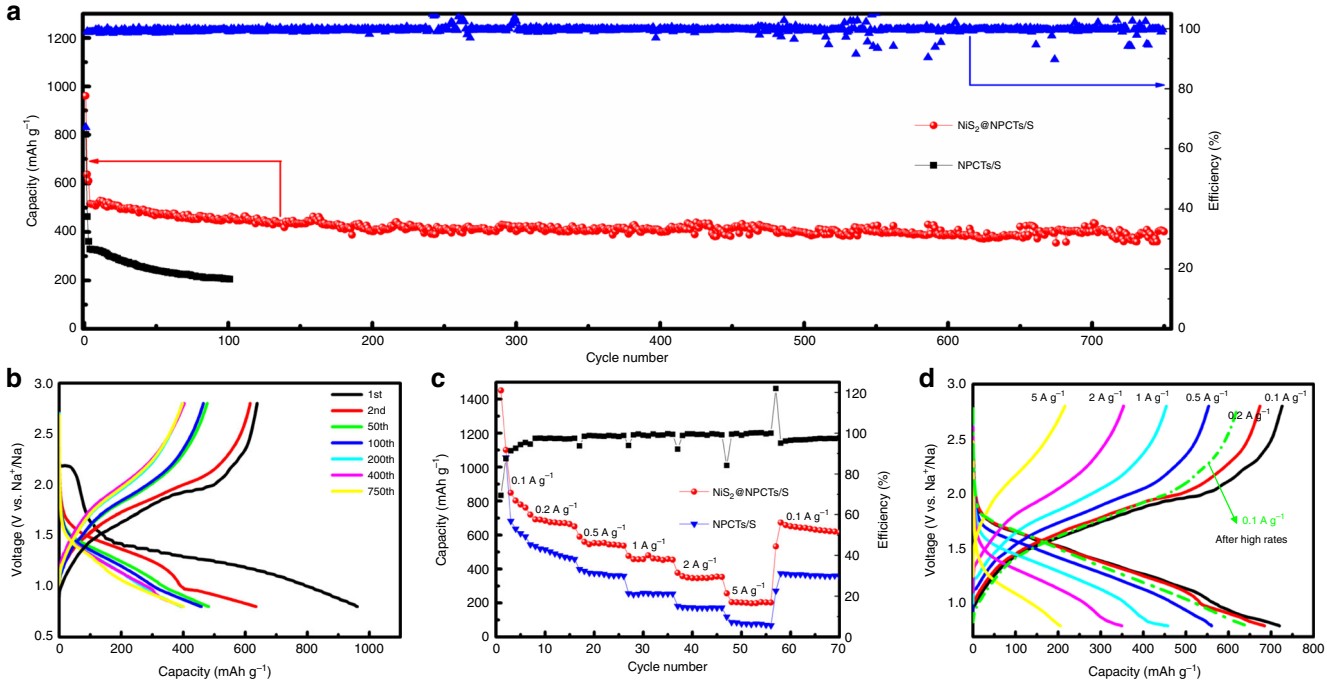

**Fig. 2** Room temperature sodium-sulfur battery test. **a** Cycling performance of $NiS_2$@NPCTs/S (red) and NPCTs/S (black) at a current density of $1\,A\,g^{-1}$. **b** The corresponding charge/discharge profiles of $NiS_2$@NPCTs/S at different cycles. **c** Rate capability at 0.1, 0.2, 0.5, 1, 2, and $5\,A\,g^{-1}$ for $NiS_2$@NPCTs/S (red) and NPCTs/S (blue). **d** The corresponding charge/discharge profiles of $NiS_2$@NPCTs/S at different current densities

**Visible adsorbability of polysulfides**. The strong polysulfide adsorption of the $NiS_2$@NPCTs is evidenced by the UV–vis spectra (Fig. 3a). The $Na_2S_6$ solutions exposed to $NiS_2$@NPCTs powder exhibit much weaker absorbance compared to the NPCTs, suggesting the effective adsorption capability of $NiS_2$ nanocrystals towards polysulfides. It is evident that the yellow $Na_2S_6$ solution turns almost transparent when exposed to $NiS_2$@NPCTs after 30 min (inset of Fig. 3a), although the color of the solution remains faint yellow for pristine NPCTs. Furthermore, optically transparent Na–S cells are shown in operation in Fig. 3b, c. After 4 h of discharging, a faint yellow color is observed in the transparent electrolyte for the NPCTs/S cell, which is due to the resultant polysulfide migration. In contrast, no obvious color change is observed for the $NiS_2$@NPCTs/S electrode. The STEM-EDS mapping images of the $NiS_2$@NPCTs/S electrode (Supplementary Fig. 11) in a sodiated state (open-circuit voltage around 0.8 V) show that the dispersion of elemental sodium and sulfur is highly overlapped, implying that all sulfur in this material is active for Na-ion storage. After 100 cycles in a desodiated state (open-circuit voltage around 2.8 V), the mapping images (Fig. 3d) show that the sulfur species have been well immobilized in the cavities and homogeneously dispersed along the carbon walls. It indicates that this hollow framework is capable of sulfur immobilization. The nitrogen-doped carbon shell with the fast electron diffusion ability and the electrocatalytic behaviors of the Lewis base matrix can provide more active sites for trapping polysulfides, which make the S species more favorable to reside in the shell of each pores during repeated charging/discharging processes. All of these observations indicate the efficient polysulfide trapping of the multifunctional $NiS_2$@NPCTs host.

**Sodium-storage mechanism**. High resolution in situ synchrotron XRD ($\lambda = 0.6687\,Å$) was carried out in RT-Na/S batteries (Fig. 4a). A peak at 10.24° for the fresh cell can be indexed to the

(222) planes of $S_8$ (JCPDS No. 77–0145). Another two peaks located at 11.55° and 13.95° are attributed to the (111) and (200) planes of $NiS_2$. During the initial discharge process, long-chain polysulfides ($Na_2S_x$) appear with three new peaks at 10.47°, 11.87°, and 12.68° when discharged to 2.0 V, indicating the solid-liquid transition from $S_8$ to long-chain polysulfides. To further understand the mechanism, $S_8$ is removed by exposing $NiS_2$@NPCTs/S composite in a 300 °C tube furnace under Ar flow for 10 mins. The XRD result (Supplementary Fig. 12a) shows only $NiS_2$ remained in this composite. However, the TGA (Supplementary Fig. 12b) shows that about 32% sulfur still remained in this composite ($NiS_2$@NPCTs/S32), indicating $S_8$ has been removed and partial sulfur exists in an amorphous state in the carbon matrix. The tested coin cell with the $NiS_2$@NPCTs/S32 composite shows that the short plateau around 2.2 V (formation of long-chain polysulfides) is no longer exist and only the plateau at 1.4 V (conversion of short-chain polysulfides) remained which resulted a high initial and reversible capacity than that of $NiS_2$@NPCTs/S composite (Supplementary Fig. 12c, d). These results indicate the plateau around 2.2 V is highly related to the reduction of $S_8$, and the amorphous sulfur remained in $NiS_2$@NPCTs/S composite can be attributed to small sulfur molecules since the electrochemical reaction start from the conversion of short-chain polysulfides[49]. Once the voltage reached 1.5 V, the $Na_2S_x$ signals faded, and a new peak at 12.82° appeared, which can be indexed to the (213) planes of $Na_2S_4$ (JCPDS No. 71–0516). A further new peak at 17.1° that emerged when the cell reached 1.25 V corresponds to the (300) planes of $Na_2S_2$ (JCPDS No. 81–1771). The intermediate $Na_2S_2$ can be further reduced to $Na_2S$ from 1.1 to 0.8 V. Two new peaks at 10.3° and 16.3° can be attributed to the (111) and (220) planes of $Na_2S$ (JCPDS No. 77–2149). More intuitive information can be observed in the contour plot of XRD patterns. The signal of $S_8$ disappeared during the charge process, indicating the irreversibility of S reduction. The signal of $Na_2S_2$ is also missing in the charge process, which might be attributed to the kinetically fast reaction.

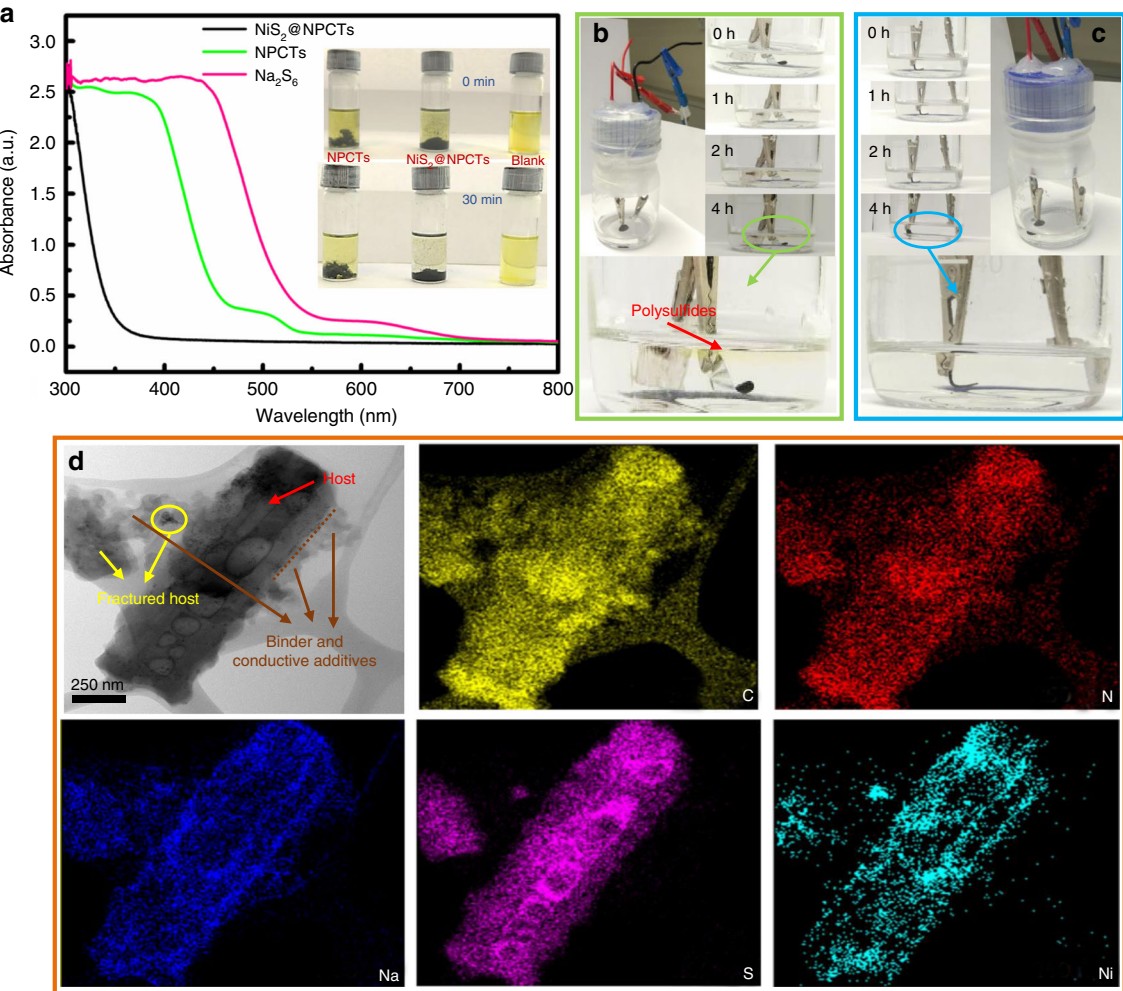

**Fig. 3** Visible adsorbability to polysulfides. **a** Ultraviolet/visible (UV–vis) spectra and corresponding photographs (inset) of pure $Na_2S_6$ solution and the solution after exposure to $NiS_2@NPCTs$ and NPCTs. Visual confirmation of polysulfide entrapment of **b** NPCTs/S and **c** $NiS_2@NPCTs$/S at specific discharge depths. **d** STEM-EDS mapping images of $NiS_2@NPCTs$/S composite after 100 cycles

This redox mechanism illustrated by in situ synchrotron XRD is consistent with the cyclic voltammograms, as clearly detailed in Supplementary Fig. 13. In general, the reversible capacity of the RT-Na/S batteries based on the $NiS_2@NPCTs$/S cathode comes from the reversibility of polysulfide conversion. The characteristic peak intensity of $NiS_2$ decreases in the region between 1.1 and 0.8 V, and recovers in the charge process. This can be related to the accumulation of $Na_2S$ and partial $Na^+$ intercalation into $NiS_2$ based on the mechanism: $NiS_2 + xNa^+ + xe^- \rightarrow Na_xNiS_2$ (details in Supplementary Fig. 8). The electrocatalytic behaviors of the N-doped sites and the $NiS_2$ component were further verified and highlighted via density functional theory (DFT) calculations. Fig 4b shows the adsorption conformations of $Na_2S_x$ on $NiS_2$ nanocrystal. The chemical interactions are dominated by the bonds between the $Na_2S_x$ and the metal sulfide (Supplementary Table 2), although there is only physical adsorption dominated by van der Waals interactions for pure carbon, which are much weaker than chemical bonds. Thus, both N-doped carbon nanotube and $NiS_2$ in our study can induce greater binding strength than pure carbon. As shown in Fig. 4c, the binding energies of $Na_2S_6$ on $NiS_2$ and N-doped carbon nanotube are 0.79 and 0.57 eV, respectively, which are much higher than on the non-doped carbon nanotube (0.09 eV), indicating their high adsorption of soluble polysulfides. More importantly, the binding energy of $Na_2S$ on $NiS_2$ is as high as 2.4 eV, which is more than

triple that on N-doped carbon. This strong binding energy of $Na_2S$ illustrates the fast reaction mechanism transforming $Na_2S_4$ into $Na_2S$. This electrocatalytic behavior can be explained by the rapid increase in binding energy via nitrogen dopant and $NiS_2$ nanocrystal. It also suggests that the dual effect of chemical binding by the nitrogen dopant and $NiS_2$ nanocrystal enables both strong entrapment of soluble polysulfides and preferential deposition of insoluble $Na_2S_2/Na_2S$ within the cathode during cycling.

## Discussion

Overall, we have developed an integrated structure to address the poor reaction kinetics of sulfur species and severe polysulfide migration. The physical confinement by the carbon shells and chemical bonding by doped nitrogen and $NiS_2$ nanocrystals are of great benefit for polysulfide immobilization. Besides, both in situ synchrotron XRD and DFT results confirm that the doped nitrogen atoms coupled with the $NiS_2$ nanocrystals serve as effective electrocatalytic sites, which significantly promote fast conversion from polysulfide to $Na_2S$. Moreover, the possible side-reaction between the dissolved polysulfide and electrolyte can be prevented by the strong polysulfide immobilization of the multifunctional sulfur host as evidenced by EDS mapping. Consequently, the novel designed cathode can deliver a high reversible

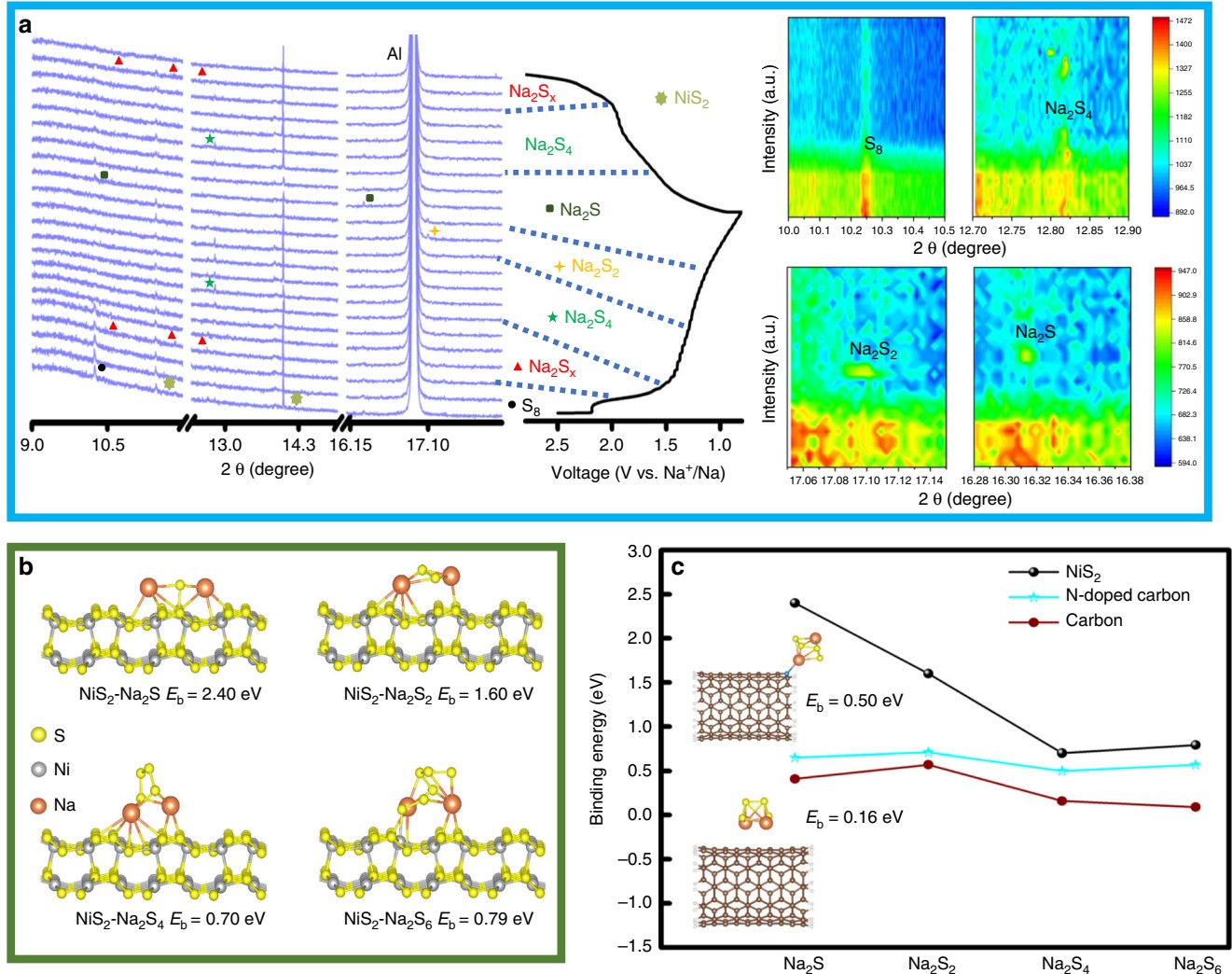

**Fig. 4** Characterization of mechanism. **a** In situ synchrotron XRD patterns of the RT-Na/S battery containing a NiS$_2$@NPCTs/S electrode with the corresponding galvanostatic charge/discharge curves at the current density of 200 mA g$^{-1}$, and contour plot of XRD patterns in selected ranges of degrees. **b** Atomic conformations and binding energies for Na$_2$S$_x$ species adsorption on NiS$_2$ (100) surface. **c** Comparison of the binding energies of various Na$_2$S$_x$ molecules bound to NiS$_2$, N-doped carbon nanotube, and carbon nanotube, respectively, with atomic conformations of Na$_2$S$_4$ adsorption on N-doped carbon nanotube and carbon as insets

capacity of 650 mA h g$^{-1}$ over 200 cycles at 0.1 A g$^{-1}$ and excellent cycling stability for 3500 cycles. Our finding on electrocatalytic polysulfide immobilization and conversion may open up a new avenue for designing diverse S-based cathodes for superior RT-Na/S batteries.

## Methods

**Synthesis of polypyrrole nanotubes**. To synthesize the polypyrrole nanotubes, Firstly, we prepared the methyl orange solution with 0.147 g methyl orange in 225 mL distilled water. After that, FeCl$_3$ (2.95 g) was added in the solution, stirring until fully dissolved. Then, the distilled pyrrole monomer ($5 \times 10^{-3}$ M) was slowly dropped in the solution with continuous stirring for overnight under room temperature. Finally, the formed polypyrrole nanotubes in the above solution were washed with distilled water and ethanol for several times.

**Synthesis of nitrogen-doped porous carbon nanotubes**. The as-prepared polypyrrole nanotubes without any pre-treatment were slightly ground in an agate mortar, then calcined at 650 °C for 5 h in Ar atmosphere to obtain the desired structure of NPCTs.

**Synthesis of the cathode composite**. The NPCTs (50 mg) were added into 50 ml deionized water, followed by ultrasonication for 3 h to form a suspension. Meanwhile, Ni(NO)$_2$·6H$_2$O (15 mg) were dissolved in 50 mL thioacetamide deionized

water solution. After stirring for 30 min, the above two solutions were mixed together, and then vigorously stirred at 50 °C. Once half deionized water evaporated, the mixed solution was stirred under vacuum at room temperature for 6 h. Then this solution was dropwise added into liquid nitrogen and freeze dried until all ice was removed. Then, the precursor was transferred to a quartz tube under Ar atmosphere and calcined at 450 °C for 5 h. The obtained composite was mixed with sulfur at weight ratio of 40/70 in a sealed quartz tube. The final NiS$_2$@NPCTs/S composite was obtained by calcine the sealed quartz tube for 155 °C for 12 h first and then 300 °C for 1 h via the facile melt-diffusion strategy. The S incorporated NPCTs (NPCTs/S) and commercial carbon nanotubes (CNT-S) were fabricated with the same conditions. Besides, the NiS$_2$@NPCTs were obtained by immersed NiS$_2$@NPCTs/S in CS$_2$ and washed for several times. Then, the material was transferred to a tube furnace under Ar atmosphere and calcined at 300 °C for 10 mins until the S has been evaporated.

**Synthesis and preparation of Na$_2$S$_6$ solution**. Eight milligram of these samples were separately immersed into 2.0 mL of 0.003 M Na$_2$S$_6$ solution in a mixed solvent of dimethoxyethane/tetraethylene glycol (DME/TEG) for 30 min.

**Physical characterization**. XRD patterns were employed with Cu Kα radiation in the 2θ range of 10°−70° (GBC MMA diffractometer, $\lambda = 1.5406$ Å, step size of 0.02° s$^{-1}$). The morphology was detected via a field emission scanning electron microscope (FESEM, JEOL JSM-7500FA) equipped with energy-dispersive X-ray spectroscopy (EDS). A 200 kV scanning transmission electron microscope (STEM,

JEM-ARM 200F) was equipped with a double aberration-corrector to achieve selected area electron diffraction (SAED) with a probe-forming, image-forming lens systems. The angular range of collected electrons for the high-angle annular dark-field (HAADF) images was around 70–250 mrad, while ABF-STEM images were recorded using a STEM-ABF detector simultaneously. The EDS mapping results were obtained via STEM using NSS software. Synchrotron powder diffraction data were collected at the Australian Synchrotron beamline with a wavelength ($\lambda$) of 0.6687 Å, calibrated with the standard reference material (National Institute of Standards and Technology (NIST) LaB6 660b). Schematic representations of the synchrotron XRD data were obtained by VESTA software. XPS with Al Kα radiation (hν = 1486.6 eV) was employed to detect the binding energies using a SPECSPHOIBOS 100 Analyser installed in a chamber in high-vacuum. The $N_2$ absorption/desorption isotherms and pore size distribution were conducted by Micromeritics Tristar 3020 analyzer (USA). Raman spectra were collected using a 10 mW helium/neon laser at 632.8 nm excitation, which was filtered by a neutral density filter to reduce the laser intensity, and a charge-coupled detector (CCD). The thermal decomposition behavior of the products was monitored by using a Mettler Toledo TGA/SDTA851 analyzer from 50 to 900 °C in Ar with a heating rate of 5 °C min$^{-1}$.

**Electrochemical measurements**. The cathode electrodes for Na–S cells which were assembled in an argon-filled glove box, were conducted by mixing 70 wt% active materials ($NiS_2$@NPCTs/S, NPCTs/S, and CNTs-S), 20 wt% carbon black, and 10 wt% carboxymethyl cellulose (CMC) binder in distilled water. The formed slurry was then pasted on Al foil via a coater (Hohsen-MC20), which was followed by drying under vacuum at 60 °C overnight. The assembled Na–S coin cells were included the punched circular working electrodes with the average mass loading of 2.5 mg cm$^{-2}$ for the active material and metallic sodium (reference and counter electrode) which were separated by glass fiber separator (Whatman GF/F). The 1 M NaClO$_4$ electrolyte used in Na–S cells were prepared by ethylene carbonate (EC)/propylene carbonate (PC) in 1:1 volume ratio, with 3 wt% fluoroethylene carbonate as additives (EC/PC + 3 wt% FEC). The electrochemical data were collected by NEWARE coin cell tester and Biologic VMP-3 electrochemical workstation with a voltage window from 0.8 to 2.8 V (vs. Na/Na$^+$).

**Computational methods**. Theoretical calculations were carried out based on the density functional theory and the plane-wave pseudopotential method[50]. The generalized gradient approximation (GGA) of the Perdew–Burke–Ernzerhof (PBE) exchange correlation function[51] was adopted with the plane-wave cut-off energy set at 500 eV. All geometry optimizations and energy calculations were performed using the periodic boundary conditions. The distance between adjacent molecules and slabs was at least 15 Å. And only Γ point was used for the reciprocal space. The criterion of convergence was set that the residual forces are less than 0.01 eV Å$^{-1}$ and the change of the total energy was <10$^{-6}$ eV. The binding energy can be expressed as $E(b) = E(Na_2S_x) + E(slab) - E(Na_2S_x@slab)$, where $E(Na_2S_x@slab)$, $E(Na_2S_x)$, and E(slab) are the total energies of the adsorbed system, the $Na_2S_x$ species, and the surface slab, respectively.

## Data availability
Data supporting the findings of this study are available from the authors on reasonable request. See author contributions for specific data sets.

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

## Acknowledgements

The authors are grateful for financial support from an Australian Renewable Energy Agency (ARENA) Project (G00849), the Innovative Group of Guangdong Province (Grant No. 2014ZT05N013), the National Natural Science Foundation of China (Grant Nos. 11704114, 61427901), and the Australian Research Council (ARC) (DE170100928).

Part of the experiments was carried out at the Powder Diffraction Beamline of the Australian Synchrotron. The authors would like to thank Guoqiang Zhao for support on TEM, Peng Li for the support on the contour plot of XRD patterns, Dr. Gilberto Casillas-Garcia for support on the STEM technique, and Dr. Tania Silver for critical reading of the paper.

## Author contributions

Z.Y. performed all synthetic experiments and prepared the manuscript. J.X. performed the density functional theory (DFT) calculations. L.W. and Hui Liu performed FTIR and UV–vis measurements and analyses. Z.Y. and F.G. conducted the TGA measurements. Z. Y., W.L., and Q.G. performed synchrotron X-ray diffraction measurements. Y.W. and S.-L.C. supervised the project. Z.Y., Y.W., S.-L.C., Hui Liu, Huakun Liu, and S.-X.D. analyzed the data and wrote the manuscript. All authors discussed the results and contributed to writing the manuscript.

## Additional information

**Competing interests:** The authors declare no competing interests.

