## [Peer Review File · Nature Communications]

Reviewers' comments:

Reviewer #1 (Remarks to the Author):

Room temperature rechargeable Na-S battery is regarded as one of the most promising alternatives to Li for next-generation energy storage systems. However, the fatal polysulfide dissolution and slow electrochemical kinetics of conversion reactions greatly impede the development of RT Na-S batteries. This work presents an intriguing multifunctional sulfur host based on NiS₂ nanocrystals implanted in nitrogen-doped porous carbon nanotubes. Benefiting from the unique structure and highly electrocatalytic activity, the as-prepared sulfur cathode with high sulfur loading delivers unprecedented rate performance at a very long cycling, although the conventional current collector and carbonate-based electrolyte are used. The characterizations regarding the structure of the cathode materials and electrochemical reaction mechanisms are well performed, showing convincing proofs for the conclusion. This synergetic strategy of the Lewis base matrix and polar surface illustrates in this manuscript would greatly boost the development of sodium-sulfur batteries. Therefore, this manuscript is recommended to be accepted for publication in Nature Communications after some minor revisions as listed below.

(1) After the sulfur loading process, why the intensity of XRD peaks is so low? Whether the encapsulation will influence its crystallinity or not.

(2) The nanotube morphology of PPy is well kept after carbonization at a high temperature. Normally, the precursor may shrink during the high temperature carbonization process. Is there any pretreatment of the PPy at low temperature?

(3) During the synthesis process, how do you control the grown of NiS₂ within the carbon tubes rather than on the exterior of NLCTs?

(4) Some related literatures are suggested to be added, like Advanced Energy Materials 2019, 9, 1803478; Advanced Functional Materials 2018, 28, 1803690; Advanced Materials 2015, 27, 7861-7866.

Reviewer #2 (Remarks to the Author):

This work reports the new findings on synthesis, structure, and electrochemical performance of NiS₂ nanocrystals@porous N-doped carbon nanotubes as cathode for room temperature rechargeable Na-S battery. The authors' viewpoint on synergized electrocatalytic effects of the Lewis base matrix and the polar nanocrystal component can effectively control the fatal polysulfide dissolution is important and interesting. And the novelty of this work on the design of well-tailored sulfur-host with physical confinement and chemical bonding for polysulfides in room temperature rechargeable Na-S battery is impressive. The authors present a comprehensive work in this manuscript, and the revealed information about this material is very appealing to the readership of the journal Nature Communications. The manuscript might be considered for publication after a minor revision.

(i) In Fig 1c, the HAADF image and the corresponding Fast Fourier Transform (FFT) pattern shows the same information, please delete, or replace it with other HRTEM image.

(ii) In Fig 3d, the STEM-EDS mapping images of NiS₂@NLCTs/S composite after 100 cycles shows a high concentration of sulfur in the edge of each pores. The author should explain why this is different with its original distribution.

(iii) As we can see the particle size of nanocrystal in this material are uniform dispersed, the authors should explain how do you control the particle size and its distribution.

(iv) The literatures in Supplementary Table 1, which the authors used as comparison in supporting information is out-of-order, please do the revision.

(v) According to Figure 3d, the mapping of sodium and nickel are highly matched. Are the reactions mostly with the sulfur absorbed on NiS₂?

Reviewer #3 (Remarks to the Author):

The work by Yan et al reports the use of NiS₂ nanocrystals supported on porous N-doped carbon nanotubes as a cathode for room-temperature Na-S batteries. The authors demonstrate that the new cathode achieves good electrochemical performance, attributing it to the polarizing surface of NiS₂ and porous structure in the conversion of soluble polysulfides into insoluble sodium sulphides, thus immobilizing them from shuttling, resulting in a high efficiency and stability. The synthetic approach and enhanced polysulfide immobilization via physical confinement and chemical bonding are interesting. The performance improvement is clear, even at a current as high as 5 A. This is a promising work towards developing room-temperature Na-S batteries. The paper is recommended for publication after the authors address the following concerns.

- The electrochemical voltage profiles of NiS₂@NLCTs/S exhibit a short plateau at 2.2 V and very rapid decrease in S redox plateau (at ~ 1.47 V during discharge), where CNTs-S shows a long plateau at 2.2 V and progressive decrease to 1.75 V. It seems that S shows a different electrochemical behaviour in NiS₂@NLCTs/S from CNTs-S. The authors need to provide more discussion regarding whether it is related to the electrocatalytic effect of NiS₂ nanocrystals.
- After the initial cycle, the capacities of CNTs-S are barely retained. Although the CNTs-S is found inactive to this system, the authors should comment on this point to give specific reasons with more careful electrochemical analysis. Why is it inactive in this system?
- A very large irreversible capacity loss is observed in the initial charge/discharge process for both samples. The authors should discuss the reason in the manuscript.
- In the discussion section: "...promoting fast conversion from polysulfide to Na₂S, preventing the active material loss from the side-reactions with the carbonate electrolyte." What do the authors mean by the "the side-reactions with the carbonate electrolyte"? Does the active material react with the carbonate electrolyte? And the DFT results cannot tell the side-reactions either.

Response to Reviewers

We would like to thank reviewers for the valuable comments and suggestions. We have revised the manuscript in accordance with the reviewers' comments and suggestions. Point-to-point responses are shown below.

Reviewer #1 (Remarks to the Author):

Room temperature rechargeable Na-S battery is regarded as one of the most promising alternatives to Li for next-generation energy storage systems. However, the fatal polysulfide dissolution and slow electrochemical kinetics of conversion reactions greatly impede the development of RT Na-S batteries. This work presents an intriguing multifunctional sulfur host based on NiS₂ nanocrystals implanted in nitrogen-doped porous carbon nanotubes. Benefiting from the unique structure and highly electrocatalytic activity, the as-prepared sulfur cathode with high sulfur loading delivers unprecedented rate performance at a very long cycling, although the conventional current collector and carbonate-based electrolyte are used. The characterizations regarding the structure of the cathode materials and electrochemical reaction mechanisms are well performed, showing convincing proofs for the conclusion. This synergetic strategy of the Lewis base matrix and polar surface illustrates in this manuscript would greatly boost the development of sodium-sulfur batteries. Therefore, this manuscript is recommended to be accepted for publication in Nature Communications after some minor revisions as listed below.

Comment 1: After the sulfur loading process, why the intensity of XRD peaks is so low? Whether the encapsulation will influence its crystallinity or not.

Response to Comment 1: Thank you very much for your comments. The lower intensity of loaded sulfur can be attributed to the reduced size and well encapsulation of the sulfur crystallites during the sulfur loading process. According to the results reported previously, it is common that sulfur encapsulated in the carbon matrix with much reduced intensity of XRD peaks compared with that of pure elemental sulfur in both lithium-sulfur and sodium-sulfur batteries. For instance, Sun et al. infused S into porous carbon matrix showed a much weaker crystalline state of the S. They speculate this phenomenon is due to the well immobilization of sulfur in carbon matrix.^[1] Besides, David Lou's group reported that the reason for the dramatic decreased intensity of well encapsulated sulfur can be attributed to the reduced size of the sulfur crystallites after the sulfur loading process.^[2] Moreover, they also declared that the low-temperature step (155°C) allowed the sulfur to melt and infuse into the carbon matrix, followed by the high-temperature step (300°C) to further promote the sulfur infusion into the central region and the pores of the carbon matrix. In that case, sulfur is well dispersed in the carbon matrix and exists in an amorphous state.^[3] We believe that the encapsulation can influence the crystallinity of sulfur. The pristine S is well crystalline (JCPDF. NO.77-0145). After

encapsulating to the NiS₂@NPCTs framework, the S peaks are less and lower intensity, which indicates that sulfur presents much reduced size and even becomes amorphous. To verify this hypothesis, we exposed NiS₂@NPCTs/S composite in a 300 °C tube furnace under Ar flow for 10 mins to remove the unencapsulated sulfur. The XRD result (Figure R1a) shows only NiS₂ remained in this composite. However, the TGA (Figure R1b) shows that about 32% sulfur still remained in this composite (NiS₂@NPCTs/S32), indicating partial sulfur exists in an amorphous state in the carbon matrix. To further understand the electrochemical behaviour of the amorphous sulfur in room-temperature Na-S battery, we assembled the coin cell with the NiS₂@NPCTs/S32 composite. It is interesting to note that the short plateau at 2.2 V (formation of long-chain polysulfides) is no longer exist and only the plateau at 1.4 V (conversion of short-chain polysulfides) remained which resulted a high initial and reversible capacity than that of NiS₂@NPCTs/S composite (Figure R1c,d). These results indicate the amorphous sulfur remained in NiS₂@NPCTs/S composite can be attributed to small sulfur molecules since the electrochemical reaction start from the conversion of short-chain polysulfides.^[4]

Figure R1 | Characterization of NiS₂@NPCTs/S32. (a) XRD patterns, (b) Thermogravimetry curve, (c) Cycling performance at a current density of 1 A g⁻¹, (d) the corresponding charge/discharge profiles.

In summary, we confirmed that the dramatic decreased intensity of well encapsulated sulfur can be attributed to the reduced size of the sulfur after loading process, and the well dispersed small sulfur molecules exist in an amorphous state.

- [1] D. Ma, Y. Li, J. Yang, H. Mi, S. Luo, L. Deng, C. Yan, M. Rauf, P. Zhang, X. Sun, X. Ren, J. Li, H. Zhang, *Advanced Functional Materials* 2018, 28, 1705537.
- [2] H. B. W. Chaofeng Zhang, Changzhou Yuan, Zaiping Guo, and Xiong Wen (David) Lou, *Angew. Chem.* 2012, 124, 1-5.
- [3] H. B. Wu, S. Wei, L. Zhang, R. Xu, H. H. Hng, X. W. Lou, *Chemistry* 2013, 19, 10804-10808.
- [4] S. Xin, Y. X. Yin, Y.-G. Guo, L. J. Wan, *Adv. Mater.* 2014, 26, 1261-1265.

We have added the detailed explanation for the low-intensity of S peaks in our revised manuscript. (see the highlight with yellow background in page 4, 9, and 10).

Supplementary Figure 12 | Characterization of NiS₂@NPCTs/S32. (a) XRD patterns, **(b)** Thermogravimetry curve, **(c)** Cycling performance at a current density of 1 A g⁻¹, **(d)** the corresponding charge/discharge profiles.

“The low-intensity S peaks of well encapsulated sulfur can be attributed to the reduced size of the sulfur after sulfur loading process, indicate the successful encapsulation of sulfur.”

“S₈ is removed by exposing NiS₂@NPCTs/S composite in a 300 °C tube furnace under Ar flow for 10 mins. The XRD result (Supplementary Fig. 12a) shows only NiS₂ remained in this composite. However, the TGA (Supplementary Fig. 12b) shows that about 32% sulfur still

remained in this composite (NiS₂@NPCTs/S32), indicating S₈ has been removed and partial sulfur exists in an amorphous state in the carbon matrix. The tested coin cell with the NiS₂@NPCTs/S32 composite shows that the short plateau around 2.2 V (formation of long-chain polysulfides) is no longer exist and only the plateau at 1.4 V (conversion of short-chain polysulfides) remained, which resulted a high initial and reversible capacity than that of NiS₂@NPCTs/S composite (Supplementary Fig. 12c,d). These results indicate the amorphous sulfur remained in NiS₂@NPCTs/S composite can be attributed to small sulfur molecules since the electrochemical reaction start from the conversion of short-chain polysulfides.”

“50. Xin, S., Yin, Y., Guo, Y.-G., Wan, L. J. A High-Energy Room-Temperature Sodium-Sulfur Battery. *Adv. Mater.* **28**, 2427 (2016).”

Comment 2: The nanotube morphology of PPy is well kept after carbonization at a high temperature. Normally, the precursor may shrink during the high temperature carbonization process. Is there any pretreatment of the PPy at low temperature?

Response to Comment 2: Thank you very much for your valuable comments. You are right about the shrinking of PPy precursor during the high temperature carbonization process. In this case, the PPy precursor shrunken and broken at 850 °C as we illustrated in Figure R2b. The nanotube morphology of PPy is well kept at 650 °C as we illustrated in Figure R2a. And we didn't do any pre-treatment of the PPy at low temperature in our synthesis. Thus, we think the precursor is stable below 650 °C.

Figure R2 | Morphology of polypyrrole nanotubes under different carbonization temperature. (a) 650 °C, (b) 850 °C.

We have cleared this point in the experiment part of our revised manuscript and Supplementary Information. (see the highlight with yellow background in page 12 and Supplementary Fig. 3).

“The as-prepared PPy nanotubes without any pre-treatment were slightly ground in an agate mortar, then calcined at 650 °C for 5 h in Ar atmosphere to obtain the desired structure of NPCTs.”

“Uniform polypyrrole (PPy) nanotubes precursor was prepared by a polymerization method. When carbonized at different temperature, N-doped carbon matrixes with distinguishing fine structures can be formed, respectively. The PPy precursor shrunken and broken at 850 °C, however, the nanotube morphology of PPy is well kept at 650 °C. The results indicate that NPCTs under 650 °C shows favourable nano-lacunose structure as a sulfur host. ”

Comment 3: During the synthesis process, how do you control the grown of NiS₂ within the carbon tubes rather than on the exterior of NPCTs?

Response to Comment 3: Thank you for your careful reading. In order to realize the NiS₂ grown within the carbon tubes, we exploited capillary effect to driven the raw materials (nickle salt and thioacetamide) into the interior pores via vacuum treatment. For comparison, a control sample was prepared by conducting the same experiment but without vacuum treatment. As shown in Figure R3a, most of the NiS₂ compounds can be visually observed by SEM without vacuum treatment, indicating the NiS₂ compounds were adsorbed on the exterior of NPCTs. However, the trace of NiS₂ compounds treated by vacuum stirring cannot be visually observed by SEM (Figure R3b), which can only be detected by TEM, indicating the NiS₂ nanocrystals grow within the carbon tubes. In addition, the following step of liquid nitrogen coupled with freeze-drying can further lock NiS₂ within the carbon tubes.

Figure R3 | SEM and TEM images of NiS₂@NPCTs/S under different synthesis conditions. (a) Stirring without vacuum, (b) stirring with vacuum.

We have added the explanation of how we control the growth of NiS₂ within the carbon tubes in our revised manuscript. (see the highlight with yellow background in page 3 and Supplementary Fig. 1)

Supplementary Figure 1 | Fabricating procedure, SEM, and HAADF-STEM images of NiS₂@NPCTs/S. SEM and TEM images of NiS₂@NPCTs/S under different synthesis conditions. (f) Stirring without vacuum, (g) stirring with vacuum.

“To realize the mechanism of NiS₂ grown within the carbon tubes, a capillary effect via vacuum treatment is introduced to drive the raw materials (nickle salt and thioacetamide) into the interior pores. For comparison, a control sample was prepared by conducting the same experiment but without vacuum treatment. As displayed in Supplementary Fig. 1f, most of the NiS₂ compounds can be visually observed by SEM without vacuum treatment, indicating the NiS₂ compounds were adsorbed on the exterior of NPCTs. However, no trace of NiS₂ compounds is observed on the surface of the NiS₂@NPCTs/S nanocomposite prepared by vacuum stirring, indicating the NiS₂ nanocrystals grow within the carbon tubes. In addition, the following step of liquid nitrogen coupled with freeze-drying can further lock NiS₂ within the carbon tubes.”

Comment 4: Some related literatures are suggested to be added, like Advanced Energy Materials 2019, 9, 1803478; Advanced Functional Materials 2018, 28, 1803690; Advanced Materials 2015, 27, 7861-7866.

Response to Comment 4: Your suggestion is greatly appreciated. We have provided the relevant reference (See references 18-20) to support the claims in our revised manuscript.

18. Wu, T., Jing, M., Zou, G., Hou, H., Zhang, Y., Cao, X., Ji, X. B. Controllable Chain - Length for Covalent Sulfur - Carbon Materials Enabling Stable and High - Capacity Sodium Storage. *Adv. Energy Mater.* **9**, 1803478. (2019)
19. Hou, H., Banks, C., Jing, M., Zhang, Y., Ji, X. B. Carbon Quantum Dots and Their Derivative 3D Porous Carbon Frameworks for Sodium - Ion Batteries with Ultralong Cycle Life. *Adv. Mater.* **27**, 7861 (2015).
20. Zhao, G., Zhang, Y., Yang, L., Jiang, Y., Zhang, Y., Hong, W., Tian, Y., Zhao, H., Hu, J., Zhou, L., Hou, H., Ji, X. B., Mai, L. Q. Nickel Chelate Derived NiS₂ Decorated with Bifunctional Carbon: An Efficient Strategy to Promote Sodium Storage Performance. *Adv. Funct. Mater.* **28**, 1803690 (2018).”

Reviewer #2 (Remarks to the Author):

This work reports the new findings on synthesis, structure, and electrochemical performance of NiS₂ nanocrystals@porous N-doped carbon nanotubes as cathode for room temperature rechargeable Na-S battery. The authors' viewpoint on synergized electrocatalytic effects of the Lewis base matrix and the polar nanocrystal component can effectively control the fatal polysulfide dissolution is important and interesting. And the novelty of this work on the design of well-tailored sulfur-host with physical confinement and chemical bonding for polysulfides in room temperature rechargeable Na-S battery is impressive. The authors present a comprehensive work in this manuscript, and the revealed information about this material is very appealing to the readership of the journal Nature Communications. The manuscript might be considered for publication after a minor revision.

Comment 1: In Fig 1c, the HAADF image and the corresponding Fast Fourier Transform (FFT) pattern shows the same information, please delete, or replace it with other HRTEM image.

Response to Comment 1: Thank you very much for your suggestions. Indeed, the HAADF image and the corresponding Fast Fourier Transform (FFT) pattern shows the same information in Fig 1c. Therefore, we have replaced the HAADF image with the corresponding HRTEM image, and revised the unnecessary explanations in our revised manuscript. (see the highlight with yellow background in page 4 and Figure 1c).

Figure 1 | Morphology, Crystal structure, Thermogravimetric, and X-ray photoelectron analysis. (a) STEM-EDS mapping images, (b) Colored STEM image coupled with EDS line scanning (inset) of a single cavity, and (c) HRTEM image with corresponding Fast Fourier Transform (FFT) pattern and molecular model matched IFFT image of the NiS₂@NPCTs/S composite (insets).

“Fig. 1c contains a high-resolution transmission electron microscopy (HRTEM) image taken on NiS₂@NPCTs/S composite shows that the interplanar distance between adjacent lattice planes is 0.279 nm, corresponding to (200) plane of NiS₂. The inset 16 formula unit crystal structure model of pyrite NiS₂ along [001] projected direction, which is highly consistent with the matched Inverse Fast Fourier Transform (IFFT) pattern, indicating a high degree of crystallinity of the NiS₂.”

Comment 2: In Fig 3d, the STEM-EDS mapping images of NiS₂@NPCTs/S composite after

100 cycles shows a high concentration of sulfur in the edge of each pores. The author should explain why this is different with its original distribution.

Response to Comment 2: Thank you very much for your valuable comments. Indeed, the sulfur distributes homogeneously throughout the hollow framework originally, and a significant quantity of the sulfur moves to the inner surface of nitrogen doped carbon shells after 100 cycles. It is indicating the ability of sulfur immobilization of this hollow framework. The nitrogen doped carbon shell with the fast electron diffusion ability and the electrocatalytic behaviors of the Lewis base matrix can provide more active sites for trapping polysulfides, which make the S species more favourable in the edge of each pores. Besides, the interior accumulation of S species along the carbon shell during each charging/discharging process for 100 cycles can inevitably increase the sulfur concentration in the edge of each pores. This phenomenon has also been found in both lithium-sulfur batteries and sodium-sulfur batteries. Li et al. reported a double-oxide sulfur host for advanced lithium-sulfur batteries, which showed high activity of the oxide shell for sulfur immobilization.^[1] And Xiao et al. has clearly illustrated the interfacial reaction between hollow carbon nanosphere and sulfur for lithium-sulfur batteries.^[2] Moreover, our early work on mesoporous carbon hollow nanospheres for high-performance room-temperature sodium-sulfur batteries has shown convincing proofs for the conclusion.^[3] As shown in Figure R3, S is favourably encapsulated in the carbon shells and embedded in the mesopores. And the carbon shell works as a wall between soluble polysulfides and electrolyte, which could greatly accumulate S species during cycling.

Figure R4 Schematic of the confinement in the S@iMCHS nanocomposite. **Copyright Ref 3**

- [1] W. Xue, Q.-B. Yan, G. Xu, L. Suo, Y. Chen, C. Wang, C.-A. Wang, J. Li, Nano Energy 2017, 38, 12-18.
- [2] J. Zheng, P. Yan, M. Gu, M. J. Wagner, K. A. Hays, J. Chen, X. Li, C. Wang, J.-G. Zhang, J. Liu and J. Xiao, Front. Energy Res. 2015, 3, 25.

[3] Y. Wang, J. Yang, W. Lai, S. Chou, Q. Gu, H. Liu, D. Zhao and S. Dou, J. Am. Chem. Soc. 2016, 138, 16576-16579.

We have added the explanation of why the S is different with its original distribution in our revised manuscript. (see the highlight with yellow background in page 8 and 9).

“After 100 cycles in a desodiated state (open-circuit voltage around 2.8 V), the mapping images (Fig. 3d) show that the sulfur species have been well immobilized in the cavities and homogeneously dispersed along the carbon walls. It indicates that this hollow framework is capable of sulfur immobilization. The nitrogen doped carbon framework with the fast electron diffusion ability and the electrocatalytic behaviors of the Lewis base matrix can provide more active sites for trapping polysulfides, which make the S species more favourable to reside in the shell of each pores during repeated charging/discharging processes.”

Comment 3: As we can see the particle size of nanocrystal in this material are uniform dispersed, the authors should explain how do you control the particle size and its distribution.

Response to Comment 3: In order to control the particle size and distribution of nanocrystals, we exploited capillary effect to driven the raw materials (nickle salt and thioacetamide) into the interior pores via vacuum treatment. For comparison, a control sample was prepared by conducting the same experiment but without vacuum treatment. As shown in Figure R5a, most of the NiS₂ compounds can be visually observed by SEM without vacuum treatment, indicating the NiS₂ compounds were adsorbed on the exterior of NPCTs. However, the trace of NiS₂ compounds treated by vacuum stirring cannot be visually observed by SEM (Figure R5b), which can only be detected by TEM, indicating the NiS₂ nanocrystals grow within the carbon tubes. In addition, the following step of liquid nitrogen coupled with freeze-drying can further lock NiS₂ within the carbon tubes, and the particle size can be effectively controlled by those pores and cavities at the same time.

Figure R5 | SEM and TEM images of NiS₂@NPCTs/S under different synthesis conditions. (a) Stirring without vacuum, (b) stirring with vacuum.

We have added the explanation of how we control the particle size and distribution in our revised manuscript. (see the highlight with yellow background in page 3, 4 and Supplementary Fig.1)

Supplementary Figure 1 | Fabricating procedure, SEM, and HAADF-STEM images of NiS₂@NPCTs/S. SEM and TEM images of NiS₂@NPCTs/S under different synthesis conditions. (f) Stirring without vacuum, (g) stirring with vacuum.

“To realize the mechanism of NiS₂ grown within the carbon tubes, a capillary effect via vacuum treatment is introduced to drive the raw materials (nickle salt and thioacetamide) into the interior pores. For comparison, a control sample was prepared by conducting the same experiment but without vacuum treatment. As displayed in Supplementary Fig. 1f, most of the NiS₂ compounds can be visually observed by SEM without vacuum treatment, indicating the

NiS₂ compounds were adsorbed on the exterior of NPCTs. However, no trace of NiS₂ compounds is observed on the surface of the NiS₂@NPCTs/S nanocomposite prepared by vacuum stirring, indicating the NiS₂ nanocrystals grow within the carbon tubes. In addition, the following step of liquid nitrogen coupled with freeze-drying can further lock NiS₂ within the carbon tubes, and the particle size can be effectively controlled by those pores and cavities at the same time.”

Comment 4: The literatures in Supplementary Table 1, which the authors used as comparison in supporting information is out-of-order, please do the revision.

Response to Comment 4: We are so sorry for the misleading, and we have re-ordered this part in the supporting information as follows. (see the highlight with yellow background in Supplementary Table 1)

Supplementary Table 1 Results from previous room-temperature sodium sulfur batteries cathode

Active material	Loading sulfur	Electrolyte	Current collector	Working current	Cycle number	Retained discharge capacity	Ref
S@C	35%	1M NaPF ₆ with 0.25M NaNO ₃ in TEGDME	Stainless steel discs	1C	1500	300mAh g ⁻¹	5
NGNS/S	25%	1M NaClO ₄ in EC:DMC:PC	Al foil	0.1C	300	66mAh g ⁻¹	6
S@iMCHS	59%	1M NaClO ₄ in EC:PC with 5%FEC	Al foil	100mA g ⁻¹	200	292mAh g ⁻¹	28
CSB@TiO ₂	60%	1M NaClO ₄ in EC:DEC	Free-standing	1A g ⁻¹ 2A g ⁻¹	1400 3000	524mAh g ⁻¹ 382mAh g ⁻¹	30
ZIF-8/S	50%	1M NaClO ₄ in TEGDME	Al foil	0.2C	250	500mAh g ⁻¹	31
c-PANS	31%	0.8M NaClO ₄ in EC:DEC	Al foil	220mA g ⁻¹	500	180mAh g ⁻¹	32
S/(CHNBs@PCNFs)	70%	1M NaClO ₄ in EC:PC with 5%FEC	Al foil	2C	400	256mAh g ⁻¹	33
MCPS1	47%	1M NaClO ₄ in EC:PC with 5%FEC	Carbon coated Al foil	0.1C	50	800mAh g ⁻¹	35
S@Con-HC	47%	1M NaClO ₄ in EC:PC with 5%FEC	Cu foil	100mA g ⁻¹	600	508mAh g ⁻¹	36
NiS ₂ @NPCTs/S	56%	1M NaClO ₄ in EC:PC with 5%FEC	Al foil	0.1A g ⁻¹ 1A g ⁻¹ 2A g ⁻¹ 5A g ⁻¹	200 1400 1800 3500	650mAh g ⁻¹ 401mAh g ⁻¹ 327mAh g ⁻¹ 208mAh g ⁻¹	This work

Comment 5: According to Figure 3d, the mapping of sodium and nickel are highly matched. Are the reactions mostly with the sulfur absorbed on NiS₂?

Response to Comment 5: Thank you for your careful reading. We've compared the mapping of sodium and nickel carefully, and labelled the different parts with yellow circles in Figure R6a. In our manuscript, the mapping image in Figure 3d is in a desodiated state (open-circuit voltage around 2.8 V), where the sodium ions were concentrated in the anode after 100 cycles. The sodium signals in Figure 3d (manuscript) can be attributed to the SEI film and residual electrolyte, rather than the sodium polysulfide which absorbed on NiS₂. Thus, it does not mean that "reactions are mostly with the sulfur absorbed on NiS₂". To clarify this point, we showed the mapping images of NiS₂@NPCTs/S electrode in a sodiated state (open-circuit voltage around 0.8 V), where the sodium ions were concentrated in the cathode. As we can see from Figure R6b, the mapping of sodium and sulfur are highly matched indicating the high activity of all sulfur in this material rather than a portion of absorbed on NiS₂.

Figure R6. STEM-EDS mapping images of the NiS₂@NPCTs/S electrode a) desodiated state, b) sodiated state.

We have added the explanation and EDS mapping image of NiS₂@NPCTs/S electrode under fully sodiated state in our revised manuscript. (see the highlight with yellow background in page 8, 9 and Supplementary Fig.11)

Supplementary Figure 11 | STEM-EDS mapping images of the NiS₂@NPCTs/S electrode under sodiated state.

“The STEM-EDS mapping images of the NiS₂@NPCTs/S electrode (Supplementary Fig. 11) in a sodiated state (open-circuit voltage around 0.8 V) show that the dispersion of elemental sodium and sulfur is highly overlapped, implying that all sulfur in this material is active for Na-ion storage. After 100 cycles in a desodiated state (open-circuit voltage around 2.8 V), the mapping images (Fig. 3d) show that the sulfur species have been well immobilized in the cavities and homogeneously dispersed along the carbon walls.”

Reviewer #3 (Remarks to the Author):

The work by Yan et al reports the use of NiS₂ nanocrystals supported on porous N-doped carbon nanotubes as a cathode for room-temperature Na-S batteries. The authors demonstrate that the new cathode achieves good electrochemical performance, attributing it to the polarizing surface of NiS₂ and porous structure in the conversion of soluble polysulfides into insoluble sodium sulphides, thus immobilizing them from shuttling, resulting in a high efficiency and stability. The synthetic approach and enhanced polysulfide immobilization via physical confinement and chemical bonding are interesting. The performance improvement is clear, even at a current as high as 5 A. This is a promising work towards developing room-temperature Na-S batteries. The paper is recommended for publication after the authors address the following concerns.

Comment 1: The electrochemical voltage profiles of NiS₂@NPCTs/S exhibit a short plateau at 2.2 V and very rapid decrease in S redox plateau (at ~ 1.47 V during discharge), where CNTs-S shows a long plateau at 2.2 V and progressive decrease to 1.75 V. It seems that S shows a different electrochemical behaviour in NiS₂@NPCTs/S from CNTs-S. The authors need to provide more discussion regarding whether it is related to the electrocatalytic effect of NiS₂ nanocrystals.

Response to Comment 1: Thank you very much for your valuable comments. Indeed, CNTs-S shows a longer plateau at 2.2 V than that of NiS₂@NPCTs/S electrode. To understand the reason for this phenomenon, we compare the XRD pattern of these two samples. The XRD pattern in Figure R7 reveals the state of sulfur in CNTs-S is very similar to the pristine S, indicating the high crystalline state and poor encapsulation in CNTs matrices. However, sulfur encapsulated in NiS₂@NPCTs/S shows much reduced intensity of XRD peaks compared with that of pure elemental sulfur (manuscript Figure 1d). As we mentioned in our manuscript, the plateau around 2.2 V is highly related to the reduction of S₈ to form long-chain polysulfides. Thus, CNTs-S electrode with more crystalline S₈ shows a stronger reduction plateau of S₈ at 2.2 V. On the contrary, NiS₂@NPCTs/S electrode with more small sulfur molecules provides a shorter plateau at 2.2 V. And that might be the reason for why S shows a different electrochemical behaviour at the initial discharge for NiS₂@NPCTs/S (with both S₈ and small sulfur molecules) and CNTs-S (only S₈). To verify those hypotheses, we exposed NiS₂@NPCTs/S composite in a 300 °C tube furnace under Ar flow for 10 mins to remove the unencapsulated S₈. The XRD result (Figure R8a) shows only NiS₂ remained in this composite. However, the TGA (Figure R8b) shows that about 32% sulfur still remained in this composite (NiS₂@NPCTs/S32), indicating partial sulfur exists in an amorphous state in the carbon matrix. To further understand the electrochemical behaviour of the amorphous sulfur in room-

temperature Na-S battery, we assembled the coin cell with the NiS₂@NPCTs/S32 composite. It is interesting to note that the short plateau at 2.2 V (formation of long-chain polysulfides) is no longer exist and only the plateau at 1.4 V (conversion of short-chain polysulfides) remained which resulted a high initial and reversible capacity than that of NiS₂@NPCTs/S composite (Figure R8c,d). These results indicate the amorphous sulfur remained in NiS₂@NPCTs/S composite can be attributed to small sulfur molecules since the electrochemical reaction start from the conversion of short-chain polysulfides at 1.4 V. Therefore, the difference in electrochemical behaviour of NiS₂@NPCTs/S and CNTs-S at initial discharge process can be attributed to the different forms of sulfur rather than the electrocatalytic effect of NiS₂ nanocrystals.

Figure R7. XRD pattern of the CNTs-S.

Figure R8. Characterization of NiS₂@NPCTs/S32. (a) XRD patterns, (b) Thermogravimetry curve, (c) Cycling performance at a current density of 1 A g⁻¹, (d) the corresponding charge/discharge profiles.

We have added the detailed discussion for the reason of different electrochemical behaviour in our revised manuscript and supporting information. (see the highlight with yellow background in page 6, 7, 9, 10 and supporting information page 10)

Supplementary Figure 9 | Cycling performance of CNTs-S. (c) XRD pattern.

Supplementary Figure 12 | Characterization of NiS₂@NPCTs/S32. (a) XRD patterns, **(b)** Thermogravimetry curve, **(c)** Cycling performance at a current density of 1 A g⁻¹, **(d)** the corresponding charge/discharge profiles.

“a commercial carbon nanotube/S mixture (CNTs-S) was compared. The CNTs-S mixture with high crystalline of S was found to be inactive (Supplementary Fig. 8 and 9).”

“To further understand the mechanism, S₈ is removed by exposing NiS₂@NPCTs/S composite in a 300 °C tube furnace under Ar flow for 10 mins. The XRD result (Supplementary Fig. 12a) shows only NiS₂ remained in this composite. However, the TGA (Supplementary Fig. 12b) shows that about 32% sulfur still remained in this composite (NiS₂@NPCTs/S32), indicating S₈ has been removed and partial sulfur exists in an amorphous state in the carbon matrix. The tested coin cell with the NiS₂@NPCTs/S32 composite shows that the short plateau around 2.2 V (formation of long-chain polysulfides) is no longer exist and only the plateau at 1.4 V (conversion of short-chain polysulfides) remained which resulted a high initial and reversible capacity than that of NiS₂@NPCTs/S composite (Supplementary Fig. 12c,d). These results indicate the plateau around 2.2 V is highly related to the reduction of S₈, and the amorphous sulfur remained in NiS₂@NPCTs/S composite can be attributed to small sulfur molecules since the electrochemical reaction start from the conversion of short-chain polysulfides.”

“Supplementary Figure 9b shows a longer plateau at 2.2 V than that of NiS₂@NPCTs/S electrode. To understand the reason for this phenomenon, A comparison XRD pattern of these two samples are discussed. The XRD pattern in Supplementary Figure 9c reveals a much stronger crystalline state of sulfur in CNTs-S, indicating the poor encapsulation in CNTs. However, sulfur encapsulated in NiS₂@NPCTs/S shows much reduced intensity of XRD peaks compared with that of pure elemental sulfur (manuscript Figure 1d). As we mentioned in our

manuscript, the plateau around 2.2 V is highly related to the reduction of S₈ to form long-chain polysulfides. Thus, CNTs-S electrode with more crystalline S₈ shows a stronger reduction plateau of S₈ at 2.2 V.”

Comment 2: After the initial cycle, the capacities of CNTs-S are barely retained. Although the CNTs-S is found inactive to this system, the authors should comment on this point to give specific reasons with more careful electrochemical analysis. Why is it inactive in this system?

Response to Comment 2: Thank you for your careful reading. As we discussed in comment 1, The XRD pattern of CNTs-S reveals a much stronger crystalline state of sulfur, indicating its poor encapsulation and the existence of S₈ in CNTs. In that case, the major reaction during the initial discharge process can be attributed to the reduction of S₈ to form long-chain polysulfides which is highly soluble in electrolyte. Thus, the severe loss of active material causes the inactive character of CNTs-S. To understand the reason why CNTs-S is inactive in this system, we further investigate the electrode process kinetics of CNTs-S and NiS₂@NPCTs/S through EIS (Figure R9). The Nyquist spectra of CNTs-S after 10 cycles shows the much bigger semicircle than that of NiS₂@NPCTs/S electrode indicating the higher charge transfer resistance of the electrode. According to the Z-view program, R_{ct} for CNTs-S is 1628 Ω which is about 9 times higher than NiS₂@NPCTs/S. Thus, we disassembled the coin cell to do further analysis. The SEM and cross-profile EDS mapping images of CNTs-S and NiS₂@NPCTs/S electrodes with digital photographs of the corresponding separator as insets are shown in Figure R10. No obvious change in the electrode and separator was observed in NiS₂@NPCTs/S electrodes. In contrast, the CNTs-S formed a thick film on the electrode surface after 10 cycles, and the separator turned brown yellow, which indicate massive active materials are dissolved in the electrolyte. Besides, the cross-profile EDS mapping images of CNTs-S shows that most of the sodium ions are concentrated on the thick film and the signal of sulfur is weak in electrode, however, sodium ions and sulfur are well dispersed all over the electrode of NiS₂@NPCTs/S. Therefore, we found the massively dissolved polysulfide in electrolyte could induce the formation of thick film on the electrode surface which greatly impeded the subsequent sodium migration during charge and discharge process.

Figure R9. Nyquist plots for CNTs-S and NiS₂@NPCTs/S composite after 10 cycles.

Figure R10. SEM, cross-profile EDS mapping images of (a) CNTs-S and (b) NiS₂@NPCTs/S electrodes with digital photographs of the corresponding separator.

In summary, the massively diffused polysulfide and formed thick film reduced the charge transfer rate and block ion accessibility, leading to slow kinetics and serious polarization, which is responsible for the inactivity of CNTs-S in Na-S system. We have added the careful electrochemical and electron microscope analyses in our revised manuscript and supporting

information. (see the highlight with yellow background in page 7 and supporting information page 10 and 11)

Supplementary Figure 9 | Characterization of CNTs-S. (d) Nyquist plots for CNTs-S and NiS₂@NPCTs/S composite after 10 cycles.

Supplementary Figure 10 | SEM, cross-profile EDS mapping images. (a) CNTs-S and (b) NiS₂@NPCTs/S electrodes with digital photographs of the corresponding separator.

“The Nyquist spectrum of CNTs-S after 10 cycles shows much higher charge transfer resistance (R_{ct}) than that of NiS₂@NPCTs/S electrode (Supplementary Fig. 9d), which is fitted to be 1628 and 207 Ω , respectively. When the cells are disassembled, the separator of CNTs-S is brown, which is ascribed to the side product of dissolved polysulfide out of CNTs framework.

In contrast, no obvious change in the electrode and separator was observed in NiS₂@NPCTs/S electrodes (Supplementary Fig. 10). Moreover, the SEM and cross-profile EDS mapping images of cycled CNTs-S electrodes show that thick film is formed on the electrode surface with dramatically reduced signal of sulfur. By contrast, uniform dispersion of S and Na is observed in NiS₂@NPCTs/S. Therefore, the severe polysulfides dissolution and formation of thick passivation film for CNTs-S lead to its failure in Na-S system.”

Comment 3: A very large irreversible capacity loss is observed in the initial charge/discharge process for both samples. The authors should discuss the reason in the manuscript.

Response to Comment 3: Thank you for your comment. Firstly, it is notable that the high-voltage plateau at 2.2 V is irreversible for both samples, indicating that the thus-formed Na₂S_x could not be reversibly oxidized to sulfur in this ethylene carbonate (EC)/propylene carbonate (PC) system, which contributes parts of the initial irreversible capacity.^[1] Secondly, partial long-chain polysulfide which produced by the solid-liquid transition during the initial discharge process could inevitably diffused in electrolyte. Moreover, just like the initial capacity loss in lithium-sulfur batteries, chemical precipitation/dissolution reactions occur during the electrochemical process resulting in active material transition between liquid phase and solid phase. But it is difficult for the high ordered sodium polysulfide to transfer completely from liquid phase to solid phase at the end of cycles, so that will lead to active material loss.^[2] Thus, polysulfide immigration and incompletely transfer reaction caused active mass loss during initial cycle can also cause irreversible capacity for both samples.

[1] Y. Wang, J. Yang, W. Lai, S. Chou, Q. Gu, H. Liu, D. Zhao and S. Dou, J. Am. Chem. Soc. 2016, 138, 16576-16579.

[2] G. Ma, Z. Wen, M. Wu, C. Shen, Q. Wang, J. Jin, X. Wu, Chem. Commun. 2014, 50, 14209-14212.

We have added the detailed discussion in our revised manuscript. (see the highlight with yellow background in page 6)

“It is notable that a large irreversible capacity loss is observed in the initial charge/discharge process for both samples, which can be attributed to the surface polysulfide dissolution and irreversible oxidization from polysulfide to sulfur.^{28, 49”}

Comment 4: In the discussion section: "...promoting fast conversion from polysulfide to Na₂S, preventing the active material loss from the side-reactions with the carbonate electrolyte." What do the authors mean by the "the side-reactions with the carbonate electrolyte"? Does the active material react with the carbonate electrolyte? And the DFT results cannot tell the side-reactions either.

Response to Comment 4: Thank you very much for your valuable comments. In the discussion part, “the side-reactions with the carbonate electrolyte” means that if the active material (polysulfide) dissolves in electrolyte, side react occurs due to the reaction of dissolved polysulfide with the used carbonate-based electrolyte, causing a thick solid electrolyte interface on the electrode. As we illustrated in Figure R10a, the CNTs-S electrode shows a brown yellow separator after 10 cycles, indicating massively diffused polysulfide in electrolyte. Consequently, the CNTs-S formed a thick film on the electrode surface, which would reduce the charge transfer rate and block ion accessibility, leading to slow kinetics and serious polarization. And that is the side-reactions we are trying to illustrate. In contrast, NiS₂@NPCTs/S electrodes with the ability of physical confinement and chemical bonding to polysulfide shows no obvious change in the electrode and separator. This phenomenon further confirmed that diffused polysulfide could react with the electrolyte. The DFT result in the discussion part aims to echo the enhanced adsorption and electrocatalytic effect of polysulfide, rather than the side-reactions. We are so sorry for the misleading, and we have rewritten this part in the discussion part. (see the highlight with yellow background in page 12)

“Besides, both in situ synchrotron XRD and DFT results confirm that the doped nitrogen atoms coupled with the NiS₂ nanocrystals serve as effective electrocatalytic sites, which significantly promote fast conversion from polysulfide to Na₂S. Moreover, the possible side-reaction between the dissolved polysulfide and electrolyte can be prevented by the strong polysulfide immobilization of the multifunctional sulfur host as evidenced by EDS mapping.”

REVIEWERS' COMMENTS:

Reviewer #1 (Remarks to the Author):

The revised version could be accepted.

Reviewer #2 (Remarks to the Author):

It could be accepted

Reviewer #3 (Remarks to the Author):

The authors have properly addressed the concerns of this reviewer from the first round review. Thus this reviewer suggests it to be accepted for publication.

Point by Point Responses:

Reviewer #1 (Remarks to the Author):

The revised version could be accepted.

Response: We highly appreciate the reviewer's positive comments.

Reviewer #2 (Remarks to the Author):

It could be accepted.

Response: We highly appreciate the reviewer's positive comments.

Reviewer #3 (Remarks to the Author):

The authors have properly addressed the concerns of this reviewer from the first round review. Thus this reviewer suggests it to be accepted for publication.

Response: We highly appreciate the reviewer's positive comments.